# Antepartum and labour-related single predictors of non-participation, dropout and lost to follow up in a randomised controlled trial comparing internet-based cognitive–behaviour therapy with treatment as usual for women with negative birth experiences and/or post-traumatic stress following childbirth

Josefin Sjömark,[1] Agneta Svanberg,[1] Frida Viirman,[1] Margareta Larsson,[1] Inger Poromaa,[1] Alkistis Skalkidou,[1] Maria Jonsson,[1] Thomas Parling [1,2]

For numbered affiliations see end of article.

**Correspondence to**
Dr Thomas Parling;
thomas.parling@kbh.uu.se

## ABSTRACT

**Objectives** Internet-based interventions are often hampered by high dropout rates. The number of individuals who decline to participate or dropout are reported, but reasons for dropout are not. Identification of barriers to participation and predictors of dropout may help improve the efficacy of internet-based clinical trials. The aim was to investigate a large number of possible predictors for non-participation and dropout in a randomised controlled trial for women with a negative birth experience and/or post-traumatic stress following childbirth.

**Setting** A childbirth clinic at a university hospital in Sweden.

**Participants** The sample included 1523 women who gave birth between September 2013 and February 2018. All women who rated an overall negative birth experience on a Likert scale, and/or had an immediate caesarean section (CS), and/or severe postpartum haemorrhage (≥ 2000 mL) were eligible.

**Methods** Demographic, antepartum, and labour-related/postpartum predictors were investigated for non-participation (eligible but denied participation), pre-treatment dropout (prior to intervention start), treatment dropout, and loss to follow-up. Descriptive statistics and logistic regression were used in the data analysis.

**Results** A majority (80.3 %) were non-participants. Non-participation was predicted by lower level of education, being foreign-born, no experience of counselling for fear of childbirth, multiparity, vaginal delivery (vs CS and vacuum-assisted delivery) and absence of: preeclampsia, anal sphincter injury and intrapartum fetal distress. Pretreatment dropout was predicted by the absence of severe haemorrhage. Treatment dropout was predicted by vaginal delivery (*vs immediate CS*), vertex presentation and good overall birth experience. Loss to follow-up was predicted by vaginal delivery (vs immediate CS or

### STRENGTHS AND LIMITATIONS OF THIS STUDY

⇒ A large number of participants from routine health-care were included
⇒ Demographic, antepartum and labour-related/post-partum predictors were investigated at four stages (recruitment, prior to treatment start, during treatment and at follow-up).
⇒ Neither psychological/psychiatric status or attitudes to internet-delivered interventions were investigated in this study but warrants further exploration.

vacuum-assisted delivery) and absence of intrapartum fetal distress.

**Conclusions** Mothers with no obstetric complications were more likely to not participate and dropout at different time points. Both demographic, antepartum and obstetrical variables are important to attend to while designing procedures to maximise participation in internet-delivered cognitive–behavioral therapy.

**Trial registration number** ISRCTN39318241

## INTRODUCTION

The internet has created new opportunities for healthcare services. Internet-delivered cognitive–behavioral therapy (iCBT) for various psychological disorders has been developed and investigated in the past decades[1] and the field is growing quickly. The active mechanisms in iCBT are the same as in CBT but differs in the way it is delivered (internet/computer based) and increases the availability for evidence based psychological interventions in the society. ICBT is

convenient, flexible and cost effective for many different psychological disorders[2]; it is effective for treatment of depression and several anxiety disorders, and for some diagnoses, iCBT is equally effective as face-to-face CBT.[3 4]

Several trials of internet interventions have had problems with high levels of non-adherence, with a majority of the participants never completing treatment.[5] Information about dropouts in internet-based interventions is generally poorly reported in the literature[5 6] and one study reported that of 75 reviewed trials, 40% failed to report information about dropouts.[7] However, when numbers are reported, they are typically high, especially in self-guided interventions.[5 8] In a review of internet-based treatments, dropout ranged between 2% and 83%, with a weighted average of 31%.[9] In a meta-analysis,[10] dropout rates of 74% were reported for unguided treatment for depression, whereas the corresponding figure for therapist-supported treatments was 28%. Kuester *et al*[11] found an average dropout rate of 23.2% in their meta-analysis of internet-based interventions for post-traumatic stress disorder (PTSD).

The literature is inconsistent regarding the definitions of participants who discontinue before treatment completion.[12] Operationalisation of adherence varies across trials and limits comparability.[13] Eysenbach[14] defines low adherence in internet interventions as *Nonuse attrition* (when a participant completes an initial assessment battery but fails to start the intervention) and *Dropout attrition* (when a participant accesses the treatment, but prematurely discontinues it). Other terms, such as "non-compliance", "failure to engage", "premature termination", "attrition" and "dropout" have been used in the literature.[12] Melville *et al*[9] identified three categories of predictors of dropout: sociodemographic factors and contextual variables, psychological problems and treatment-related variables—and described that dropout could occur at several different timepoints in iCBT. The following terms for dropout at different timepoints in internet interventions have been suggested: (1) pretreatment dropout: when a participant drops out before starting the intervention. (2) Treatment dropout: when a participant drops out after having started the intervention. (3) Follow-up dropout: when a participant completes the intervention but drops out before follow-up measures are completed.

Studies seldom report reasons for non-participation or dropout.[15] To better understand who will benefit from internet-based interventions and improve usability and efficacy, there is a need to identify factors related to dropout.[16] Adherence to internet interventions can be influenced by several sociodemographic factors, such as gender, age and level of education.[9 16–19] In a study, 96 adult patients with post-traumatic stress reactions were allocated to 10 sessions of iCBT or to a waiting list. The dropout rate in the iCBT group was 16%; technical problems and emotional distress due to the treatment interventions were the most frequently reported dropout reasons.[20]

The form of an intervention differs in internet treatments, considering amount of material, intensity and support. Some interventions are, for example, fully therapist supported with face-to-face sessions or via phone, some offer support via mail, and some do not offer support at all (self-help).[2] Systematic reviews have found that guided internet treatments in general tend to be more effective than non-guided ones.[8] Studies seldom report data on the invited persons who decline participation (non-participants). In a randomised controlled trial (RCT) investigating expressive writing for postpartum physical and psychological health, recruitment was low (10.7% of the invited).[21] The recruited sample derived from a restricted sociodemographic range (high proportion of white Europeans, well-educated, employed, many in professional occupations, older and more likely to be married).

About 115 000 women give birth in Sweden every year.[22] Childbirth is a subjective and multidimensional event that in some cases can lead to a negative childbirth experience. The prevalence of negative childbirth experiences varies (9–45%) in different communities.[23–25] For some women (3%–4%), the distress of a negative childbirth experience lead to the development of Posttraumatic Stress Disorder Following Childbirth (PTSD FC).[26–31] In Sweden, there is no specific treatment recommendation for women with negative birth experiences and/or PTSD FC. So far, for only a few RCTs, the efficacy of different interventions for this population has been investigated; there is therefore no or little information about how women with negative birth experiences commit and engage in iCBT and similar treatments.

The aim of this study was to investigate a number of possible predictors for non-participation and dropout in an RCT for those with a negative birth experience and/or post-traumatic stress following childbirth.[32] The main objective was to investigate demographic, antepartum and labour-/postpartum-related predictors for the following events: (1) non-participation (eligible women who did not give written consent), (2) pretreatment dropout (ie, dropout prior to intervention, but after having given informed consent), (3) treatment dropout (ie, dropout during treatment), and (4) loss to follow-up (ie, those who did not complete follow-up measures).

## METHODS
The Strengthening the Reporting of Observational Studies in Epidemiology cohort reporting guidelines were used for this publication.[33]

### Patient and public involvement
Patients or the public were not involved in the design, or conduct, or reporting, or dissemination plans of this study.

**Table 1** Demographics for the eligible participants (total sample) consisting of those who participated and the non-participants

| | Total n=1523 n (%) | Participants n=300 n (%) | Non-participants n=1223 n (%) |
|---|---|---|---|
| Relationship status | | | |
| Married/cohabit | 1291 (95.1) | 286 (21.1) | 1005 (74.1) |
| Single/other | 66 (4.9) | 8 (0.6) | 58 (4.3) |
| Education | | | |
| Elementary school | 72 (5.4) | 1 (0.1) | 71 (5.3) |
| High school | 489 (36.6) | 82 (6.1) | 407 (30.5) |
| University | 775 (58.0) | 209 (54.2) | 566 (42.4) |
| Country of birth | | | |
| Sweden | 953 (76.7) | 261 (21) | 692 (55.7) |
| Foreign born | 289 (23.3) | 25 (2) | 264 (21.3) |

Note: missing data; n=166 for relationship status, n=187 for education, and n=281 for country of birth.

## Study design

Investigation of single predictors for non-participation, pretreatment dropout, treatment dropout and loss to follow-up, reflecting four consecutive time points (about 8 weeks postpartum, about 10 weeks post partum, between 10 and 16 weeks post partum, and after 16 weeks post partum, respectively), for all eligible participants in a longitudinal RCT.

## Participants

The current study is a secondary analysis of an RCT for women with negative birth experiences, recruited in routine public healthcare. Approximately, 17 000 women gave birth at Uppsala university Hospital between September 2013 and February 2018, and most of them rated their overall birth experience on a Likert scale (0–10), as a standard procedure before hospital discharge. Eligible women (n=1523) had a negative birth experience (defined as≤5 on the Likert scale), and/or an immediate caesarean section, and/or a severe postpartum haemorrhage (≥ 2000 mL). Of 1523 eligible women, about 20% (n=300) gave written consent to be part of the RCT.[32] The 1523 eligible women had a mean age of 31.5 years (*SD*=5.03), participants in the RCT study were 31.7 (4.6) years and the non-participants age were 31.4 (5.1) years; the majority reported being married or having a partner (84.6%, n=1291) and 50.8% (n=775) had a university degree. Data on eligible participants are presented in table 1.

## Sample size and power

There was no specific sample size calculation for this investigation other than the sample size estimation for the RCT[21] (power was set to 0.8 with a medium effect size) where a total sample size of 130 was needed.

## Procedure

Women rated their birth experience as a routine measure at the hospital before discharge. Those with negative birth experiences were contacted via telephone, about 8 weeks post partum. During the telephone calls, the women were informed about the study and those interested in participating were sent study information and a consent form by post. Those who declined at this stage (n=693) were asked about their reason for doing so. In total, 530 eligible women did not respond to the invitation, 300 women gave written consent (participants) and 1223 did not (non-participants). Of the 300 participants, 101 never completed baseline measures (pretreatment dropouts). The participants who filled out the baseline questionnaires (n=199) were randomised to either treatment as usual (TAU, n=100) or iCBT+TAU (n=99). The iCBT treatment consisted of six treatment modules including psychoeducation and interventions, with therapist support on demand, tailored for women with negative experiences of childbirth (see online supplemental table 1).[21] Regardless of treatment allocation, local healthcare providers in accordance with international guidelines treated all participants in the study. TAU included conventional support in accordance with the existing practices at the Department of Obstetrics and Gynecology of the participating hospital. Of the 99 allocated to treatment, a total of 41 were treatment completers (at least three of six steps completed) and 58 were treatment dropouts. All randomised participants (199) were asked to fill out questionnaires 6 weeks post randomisation; 121 completed the follow-up measures and 78 were lost to follow-up; see figure 1.

## MATERIAL

Based on previous knowledge about possible causes for non-participation and dropout, predictor variables were categorised into three conceptual categories (demographic, antepartum, and labour-/postpartum-related variables). Obstetric data were extracted from each participant's medical records and questionnaire information

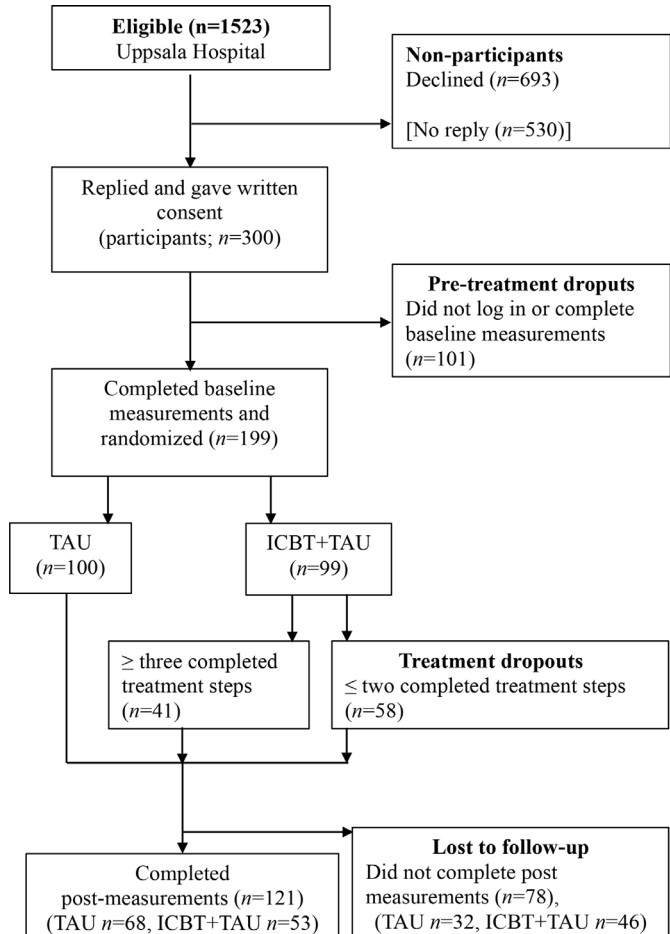

**Figure 1** Flow chart.

was taken from the U-CARE database. The Care Base Internet Platform, including its web-based part (U-CARE eService), was developed within the U-CARE programme. The aim of the U-CARE research programme is to prevent and reduce psychosocial malfunctioning in patients and relatives. The U-CARE eService is currently being used for interventions and data collection http://www. u-care. uu.se.

### Demographic variables
Country of birth (born in Sweden /foreign-born), level of education (university/high school/elementary school), relationship status (married/partner or single/other status), and age at delivery (years).

### Antepartum variables
Previous caesarean section (no/yes), counselling for fear of childbirth (no/yes), preeclampsia during pregnancy (International Classification of Disease 10th revision (ICD-10 code O14; no/yes), length of pregnancy (number of days based on second trimester ultrasound), and parity (first child/second child/third child or more).

### Labour-related/postpartum variables
Mode of delivery (vaginal delivery/emergency caesarean section/immediate caesarean section/vacuum-assisted delivery/elective caesarean section), fetal presentation (vertex presentation/others), manual placenta removal (ICD-10 code O73; no/yes), epidural anaesthesia (ICD-10 code ZXH50; no/yes), intrapartum fetal distress (ICD-10 code O68; no/yes), anal sphincter injury (ICD-10 code O70; no/yes), labour dystocia (ICD-10 code O62; no/yes), severe postpartum haemorrhage (≥ 2000 mL; no/yes), anaemia (ICD-10 code D59; no/yes), blood transfusion (ICD-10 code Z51.3; no/yes), number of children in the pregnancy (one/two or more), child transferred to neonatal intensive care unit (no/yes), breastfeeding problems (ICD-10 code O92; no/yes) and overall birth experience (more severe 0–2 vs less severe 3–5).

### Dependent variables
Non-participation (n=1223) versus participation (n=300): eligible women who did not/did return a signed informed consent form.

Pretreatment dropouts (n=101) versus pretreatment completers (n=199): women who gave written consent but did not/did proceed to complete the baseline measurements.

Treatment dropouts (n=58) versus treatment completers (n=41): women in the iCBT arm who reported activity in 0–2 treatment steps of the treatment versus 3–6 treatment steps.

Lost to follow-up (n=78) versus completed follow-up: all randomised women in either treatment arm (iCBT+TAU and TAU) who never completed the post-treatment measures versus those who completed them (n=121).

### STATISTICAL ANALYSIS
Logistic regression was used to determine predictors of non-participation, pretreatment dropout, treatment dropout and loss to follow-up. ORs with 95% CIs and beta values including SE are reported. Missing data were not handled. Among the predictor variables, there were three demographics that had missing data above 5%; country of birth, education, and relationship status (18.5%, 12% and 10.8% missing, respectively). We decided not to impute these demographic missing data from antepartum, and labour-related/postpartum variables, as we think it would have resulted in arbitrary imputations. In addition, since the analyses are not multivariate but bivariate the consequences are less. All available data were used leading to slight differences regarding number of participants in the analyses. Reasons given for non-participation were categorised. SPSS V.26 was used for all analyses.

### RESULTS
#### Predictors of non-participation
Women with lower levels of education, multiparas and foreign born were more often non-participants (table 2). Women who had not been counselled for fear of childbirth, no preeclampsia during pregnancy, no sphincter injury, no intrapartum fetal distress, and those with

**Table 2** OR with 95% CI derived from logistic regression for potential predictors

| | Non-participants | | Pretreatment dropout | | Treatment dropout | | Lost to follow-up | |
|---|---|---|---|---|---|---|---|---|
| | N | OR 95% CI | N | OR 95% CI | N | OR 95% CI | N | OR 95% CI |
| Country of birth | 1234 | 3.93 (2.58 to 6.15)*** | 284 | 1.09 (0.45 to 2.64) | 98 | 1.43 (0.33 to 6.06) | 198 | 1.87 (0.69 to 5.08) |
| Sweden/other | 953/281 | | 259/25 | | 89/9 | | 181/17 | |
| Level of education | 1336 | | 290 | | 99 | | 199 | |
| University | 775 | 1.0 | 208 | 1.0 | 71 | 1.0 | 148 | 1.0 |
| High school | 489 | 1.83 (1.38 to 2.44)*** | 81 | 1.53 (0.89 to 2.62) | 27 | 1.32 (0.53 to 3.28) | 50 | 1.33 (0.69 to 2.55) |
| Elementary school | 72 | 26.2 (3.62 to 189.9)*** | † | na | † | na | † | na |
| Relationship status | 1357 | | 292 | | 97 | | 197 | |
| Married–cohabit/other | 1291/66 | 2.06 (0.97 to 4.37) | | 1.25 (0.29 to 5.35) | 95/2 | na | | na |
| Age, years | 1510 | 0.99 (0.96 to 1.01) | 300 | 0.97 (0.92 to 1.02) | 99 | 0.95 (0.86 to 1.05) | 199 | 0.95 (0.89 to 1.02) |
| Previous CS yes/no no/yes | 1302/189 | 0.80 (0.53 to 1.19) | 31/264 | 0.78 (0.36 to 1.68) | 89/9 | 1.42 (0.33 to 6.06) | 177/19 | 0.87 (0.33 to 2.32) – |
| Counselling for fear of childbirth, no/yes | 1523 | | 300 | | 99 | | 199 | |
| Childbirth, no/yes | 1376/147 | 0.50 (0.35 to 0.73)*** | 254/46 | 1.06 (0.55 to 2.05) | 87/12 | 0.68 (0.18 to 2.52) | 169/30 | 0.88 (0.39 to 1.97) |
| Preeclampsia, no/yes | 1440/83 | 0.59 (0.36 to 0.96)* | 276/24 | 1.20 (0.51 to 2.85) | 90/9 | 1.46 (0.34 to 6.22) | 184/15 | 1.39 (0.48 to 4.00) |
| Pregnancy, days | 1507 | 1.00 (0.99 to 1.00) | 299 | 0.99 (0.98 to 1.01) | 99 | 1.01 (0.98 to 1.04) | 198 | 1.00 (0.98 to 1.02) |
| Parity | 1505 | | 299 | | 99 | | 198 | |
| First child | 838 | 1 | 200 | 1 | 68 | 1 | 134 | 1 |
| Second child | 448 | 1.65 (1.22 to 2.22)** | 71 | 0.97 (0.55 to 1.73) | 22 | 1.80 (0.65 to 4.96) | 48 | 1.07 (0.54 to 2.10) |
| Third child or more | 219 | 2.15 (1.40 to 3.30)*** | 28 | 1.52 (0.68 to 3.40) | 9 | 1.68 (0.39 to 7.26) | 16 | 1.63 (0.58 to 4.60) |
| Mode of delivery | 1523 | | 300 | | 99 | | 199 | |
| Vaginal delivery | 783 | 1 | 129 | 1 | 40 | 1 | 82 | 1 |
| Emergency CS | 289 | 0.71 (0.51 to 1.0) | 63 | 1.0 (0.54 to 1.88) | 20 | 0.33 (0.11 to 1.03) | 40 | 0.74 (0.34 to 1.58) |
| Immediate CS† | 186 | 0.54 (0.37 to 0.79)** | 49 | 0.63 (0.30 to 1.30) | 19 | 0.19 (0.06 to 0.63)** | 36 | 0.42 (0.18 to 0.99)* |
| Vacuum assisted | 198 | 0.62 (0.43 to 0.91)* | 48 | 0.96 (0.48 to 1.91) | 15 | 0.29 (0.84 to 1.01) | 31 | 0.26 (0.10 to 0.71)** |
| Elective CS | 67 | 1.01 (0.52 to 1.99) | 11 | 0.17 (0.02 to 1.41) | 5 | 1.33 (0.13 to 13.37) | 10 | 2.57 (0.62 to 10.65) |
| Fetal presentation | 1510 | | 300 | | 99 | | 199 | |
| Vertex/other | 1287/223 | 1.02 (0.71 to 1.46) | 256/44 | 0.53 (0.25 to 1.13) | 82/17 | 0.31 (0.11 to 0.94)* | 165/34 | 1.28 (0.61 to 2.70) |
| Manual placenta removal | 1523 | | 300 | | 99 | | 199 | |
| No/yes | 1379/144 | 1.41 (0.88 to 2.26) | 278/22 | 0.42 (0.14 to 1.26) | 93/6 | 0.33 (0.06 to 1.90) | 181/18 | 0.99 (0.36 to 2.66) |
| Epidural anaesthesia | 1523 | | 300 | | 99 | | 199 | |
| No/yes | 813/710 | 0.86 (0.67 to 1.10) | 151/149 | 1.12 (0.69 to 1.80) | 49/50 | 0.95 (0.43 to 2.12) | 102/97 | 1.52 (0.86 to 2.70) |

**Table 2** Continued

| | Non-participants | | Pretreatment dropout | | Treatment dropout | | Lost to follow-up | |
|---|---|---|---|---|---|---|---|---|
| | N | OR 95% CI | N | OR 95% CI | N | OR 95% CI | N | OR 95% CI |
| Intrapartum fetal distress | 1523 | | 300 | | 99 | | 199 | |
| No/yes | 1234/289 | 0.67 (0.50 to 0.91)* | 228/72 | 0.76 (0.43 to 1.36) | 71/28 | 0.50 (0.21 to 1.21) | 148/51 | 0.50 (0.25 to 0.99)* |
| Anal sphincter injury | 1523 | | 300 | | 99 | | 199 | |
| no / yes | 1447/76 | 0.48 (0.29 to 0.79)** | 275/25 | 0.92 (0.38 to 2.21) | 88/11 | 1.27 (0.35 to 4.66) | 182/17 | 1.09 (0.40 to 3.00) |
| Labour dystocia | 1523 | | 300 | | 99 | | 199 | |
| no / yes | 885/638 | 0.87 (0.67 to 1.11) | 166/134 | 1.26 (0.78 to 2.04) | 55/44 | 0.74 (0.33 to 1.66) | 114/85 | 0.75 (0.42 to 1.34) |
| Severe haemorrhage† | 1523 | | 300 | | 99 | | 199 | |
| no / yes | 1329/194 | 1.19 (0.80 to 1.76) | 266/34 | 0.38 (0.15 to 0.96)* | 83/16 | 0.49 (0.17 to 1.44) | 171/28 | 1.00 (0.44 to 2.28) |
| Anaemia | 1523 | | 300 | | 99 | | 199 | |
| no / yes | 1274/249 | 0.85 (0.61 to 1.18) | 246/54 | 0.57 (0.29 to 1.12) | 79/20 | 0.50 (0.19 to 1.35) | 158/41 | 0.66 (0.32 to 1.38) |
| Blood transfusion | 1523 | | 300 | | 99 | | 199 | |
| no / yes | 1340/183 | 1.01 (0.69 to 1.49) | 265/35 | 0.65 (0.29 to 1.45) | 87/12 | 0.31 (0.09 to 1.09) | 173/26 | 0.53 (0.21 to 1.32) |
| Children in the pregnancy | 1508 | | 299 | | 99 | | 198 | |
| 1 child / 2 children | 1476/32 | 1.35 (0.52 to 3.55) | 294/5 | 0.48 (0.05 to 4.40) | 97/2 | na | 194/4 | 0.51 (0.05 to 4.96) |
| Child transferred to NICU | 1523 | | 300 | | 99 | | 199 | |
| no / yes | 1255/268 | 0.83 (0.60 to 1.14) | 241/59 | 1.21 (0.67 to 2.20) | 78/21 | 0.73 (0.28 to 1.91) | 162/37 | 1.07 (0.52 to 2.22) |
| Breastfeeding problems | 1523 | | 300 | | 99 | | 199 | |
| no / yes | 1505/18 | 0.86 (0.28 to 2.64) | 296/4 | 0.65 (0.07 to 6.36) | 98/1 | na | 196/3 | 0.77 (0.07 to 8.67) |
| Overall birth experience† | 1203 | | 234 58/176 58/176 | | 72 | | 148 | |
| 0–2/3–5 | 305/898 | 1.02 (0.74 to 1.42) | 58/176 | 0.72 (0.38 to 1.35) | 20/52 | 0.27 (0.09 to 0.79)* | 40/108 | 0.90 (0.43 to 1.88) |

Note: the first category is the reference, for example, when yes/no is stated, yes is the reference category.
*p<.05, **p<.01, ***p<.001.
†inclusion criteria.

**Table 3** Reasons for non-participation (n=693) given during telephone interview 8 weeks post partum

| Reason for non-participation | N | % |
|---|---|---|
| Feels fine, does not need any support | 326 | (47) |
| Does not speak Swedish | 134 | (19) |
| Not interested (no further information) | 77 | (11) |
| Feels fine, has already received professional support | 35 | (5) |
| Does not feel fine, receiving/waiting for other professional support | 35 | (5) |
| Does not have the time | 30 | (4) |
| Does not have a computer | 16 | (2) |
| Not interested, will not have more kids anyway | 14 | (2) |
| Not interested, does not want to think about the delivery | 13 | (2) |
| Misunderstood the Likert scale (inclusion), had a positive experience | 10 | (1.4) |
| Not comfortable with internet/computer, prefers face-to-face therapy | 3 | (0.4) |

vaginal delivery were more likely to decline participation, see table 2 (and online supplemental table 2 for ß,SE).

### Reasons why women declined to take part despite eligibility
Of the contacted women, 693 actively declined participation and their answers were categorised into different subgroups (table 3).

### Predictors of pretreatment dropout
The only significant predictor of pretreatment dropout was no severe postpartum haemorrhage (ie, less than 2000 mL; table 2).

### Predictors of treatment dropout
For those randomised to the treatment group, dropout was significantly predicted by mode of delivery, fetal presentation and overall birth experience (table 2). Participants with vaginal delivery, vertex presentation and less severe overall birth experience were more likely to dropout from treatment.

### Predictors of loss to follow-up
In the analyses of loss to follow-up, absence of intrapartum fetal distress and vaginal delivery (compared with immediate CS and vacuum delivery) predicted loss to follow-up (table 2). An additional analysis showed that being randomised to iCBT+TAU was a significant predictor of loss to follow-up OR=1.84 (95 % CI 1.04 to 3.28), *B*=0.61, *SE*=0.29, p=0.037, where 46 of 99 in iCBT+TAU and 32 of 100 in TAU were lost to follow-up.

### DISCUSSION
The current study provides an explorative analysis of predictors for non-participation and dropout at different

timepoints in an RCT examining iCBT for women with negative birth experiences and/or post-traumatic stress following childbirth.[21] Significant predictors for non-participation and dropout were found at different stages in the recruitment process of an RCT. Women with higher education level, without previous children and those born in Sweden were more likely to enter into the study. Thereafter, women who had been counselled for fear of birth, experienced complications during the childbirth and with an overall severe birth experience were more likely to stay in the study.

A majority (80.3%) of the eligible women declined participation and our first conclusion was that a large number of those eligible did not see themselves as being in need of iCBT or wanted to take part in a clinical trial. When they were contacted by telephone during the recruitment period, the most frequent reason for declining was 'I feel fine/have no need of any support'. Explanations could be that the cut-off for the screening instrument was overinclusive and/or that the other inclusion criteria (immediate caesarean section and severe postpartum haemorrhage) did not necessarily result in a negative birth experience. The content validity of the one-item Likert scale in the current trial could be discussed, as women may take different aspects of their birth experience into account. Also, the timepoint for the rating could be discussed. In the current trial, all women rated their birth experience shortly after giving birth; it is difficult to determine the timepoint that would yield the most accurate rating of the birth experience. However, using a Likert scale as a tool for self-assessment of overall birth experience is well-established in clinical practice and used in research.[34 35] A person's perception of their birth experience can change over time and it is important to consider the specific timepoint used in measurement.[33] Larsson *et al*[36] used a VAS scale (range 1–10) for self-assessment of birth experience at 2 days, 3 months and 9 months post partum. They found that the participants' negative birth experiences decreased over time and suggested that use of a VAS scale was an adequate way to find women in need of follow-up after a negative birth experience.

The analysis included the inclusion criteria (immediate caesarean section, overall birth experience, and severe haemorrhage) as predictors. Non-participation was predicted by vaginal delivery vs immediate caesarean section. Childbirth without severe haemorrhage predicted pretreatment dropout. It is known that severe postpartum haemorrhage is a significant risk factor for developing PTSD.[37 38] Treatment dropout was predicted by a less severe overall birth experience and vaginal delivery vs immediate CS. These predictors were inclusion criteria and must therefore be interpreted with caution. However, the results might be useful for future hypothesis in further research. The three inclusion criteria in this study are experiences that potentially can have serious effects on the mental health of a birth giving woman. It may be of value to understand more about what type of

care (eg, counselling, therapy), what type of format (eg, face to face or ICBT) and what level of support (therapist support or pure self-help) is demanded.

Three sociodemographic variables predicted non-participation: lower level of education, multiparity and being foreign born. Lower level of education as a greater risk for dropout is consistent with dropout in other iCBT trials.[16] Multiparity was also identified as an important predictor of non-participation. The physical and psychological changes of the postpartum period are challenging for first-time mothers and they have lower levels of maternal confidence and higher levels of stress compared with multiparous women,[39] which might increase their likelihood of participation. An alternative explanation is that multiparous women have less time to commit to clinical trials compared with first-time mothers. This trial's intervention addressed Swedish-speaking women; foreign-born women might see language as a barrier to participation.

Five antepartum and labour-related/postpartum variables also predicted non-participation. Women without experience of the following were more likely to be non-participants; counselling for fear of childbirth, preeclampsia anal sphincter injury, intrapartum fetal distress, and vacuum-assisted delivery (vs vaginal delivery). Women who had been counselled for fear of childbirth had already professionally addressed peripartum psychological problems and might therefore have been more open to support. Preeclampsia, intrapartum fetal distress and anal sphincter injuries are all severe conditions and motivators for participation. Preeclampsia and severe postpartum haemorrhage are significant threats to the mother and may have devastating or lethal outcomes. Further, both preeclampsia and intrapartum fetal distress are potential threats to the health of the fetus, thus acting as significant stressors for the woman. Instrumental deliveries may be caused by emergency obstetric complications potentially threatening the mother or the child and are very stressful situations for the woman in their own right.[30] The labour-related/postpartum predictors show a consistent pattern where women who did not experience these stressful events may not have had enough motivation to seek out help or support.

Pretreatment dropout was predicted by the absence of severe haemorrhage which is mentioned above. Vertex fetal presentation (vs other presentation) predicted treatment dropout. This is consistent with the significant predictors for non-participation where those with vertex presentation might not experience this as a stressful event enough to stay in the treatment. It might also be that those with vertex presentation who were randomised to the treatment did not find it helpful or that it did not address their problem fully in order to stay in the treatment.

Predictors for loss to follow-up were vaginal delivery versus instrumental delivery and absence of intrapartum fetal distress. Occurrence of such events are threats to the fetus which in turn can be very stressful for the mother. Absence of these events might lead to lost interest in

devoting time and energy to proceed with the follow-up assessment. Absence of immediate CS was also a significant predictor of loss to follow-up and is discussed in relation to the other inclusion criteria/predictors. Finally, randomisation to iCBT+TAU (compared with TAU) was a significant predictor of lost to follow-up. A majority of those who were lost to follow-up from the iCBT+TAU group were also those who were treatment droputs.

Closer inspection of variables associated with non-participation and dropout can yield insights that can be used in both future research and clinical practice. Knowledge of subgroups that are more likely to continue and complete study participation provides information about motivational factors and should be applied during the initial recruitment for similar trials. Participants where not asked why they dropped out. We believe that dropout can depend on different factors such as lack of energy and/or interest of being part of a clinical trial; dropout can also depend on the participant's experience of not needing the intervention anymore. In this trial, women with lover level of education, multipara and foreign born where more often non-participants, perhaps the way of inclusion and the intervention itself must be better adapted to attract those subgroups in the future. Translation to other languages, using simple language and pictorial material could be ways of improving adherence.

Analyses of predictors of non-participation and dropout are important for evaluating the efficacy of the interventions.[6] This explorative study found predictors of non-participation and dropout that should be taken into account in future development of similar interventions. Awareness of characteristics among women who dropout and those who continue, and complete interventions is important and should get more attention during initial recruitment for similar trials.

### Strengths
The current study is the first to present data on non-participation and dropouts in iCBT for women with negative birth experiences and/or posttraumatic stress following childbirth. The main strength of this study was the size of the sample and the routine public healthcare setting as well as consecutive recruitment. All women who gave birth were asked to rate their birth experience on a self-assessed Likert-scale and all women with a low rating were invited. This process increased the likelihood of the results being generalisable to similar clinical contexts. The exploration of dropout predictors from a large cluster of demographic variables and medical/clinical characteristics was another strength. A third strength was that reasons for dropout were explored at different stages in the study process, which allowed analysis of specific timepoints when participants were more likely to end their participation. Analyses of different timepoints for dropout could simplify the analyses of underlying reasons for withdrawal.[9]

## Limitations

Our study has several limitations that should be noted. Psychological problems and/or treatment-related variables were not available for analyses. Such variables are likely to be strong predictors of non-participation and dropout[9] and should be integrated in future studies. Neither discomfort with the internet or computers were analysed as factors for non-participation or dropout in the current trial. The impact of computer-related factors on adherence has been described previously.[17] Further, recruitment to the study was before discharge from the hospital when the experience of birth is fresh. Thus, the eligible sample might have been different if we had asked them at a later time point. However, the assessment of the birth experience was in close conjunction to immediate CS and severe haemorrhage (the two other inclusion criteria). In some analyses, there might have been a lack of power, due to the varying N, that prevented significant predictors to be found.

## CONCLUSIONS

In this sample, drawn from a large population, predictors were found for non-participation and dropout at different stages in the recruitment process and during the study of an RCT. In summary, both demographic and obstetrical variables are important to attend to for both clinical and research purposes, while designing procedures to maximise participation in iCBT for postpartum women. First time mothers with high level of education and those who had adverse obstetric experiences were more likely to join and stay in the internet intervention.

**Author affiliations**
¹Department of Women's and Children's Health, Uppsala Universitet, Uppsala, Sweden
²Centre for Psychiatry Research, Department of Clinical Neuroscience, Karolinska Institutet & Stockholm Health Care Services Region Stockholm, Stockholm, Sweden

**Acknowledgements** We would like to thank statistician Per Wikman for help with structuring the database.

**Contributors** Writing - original draft (JS), writing - review and editing (all authors), conceptualisation (ASvanberg, ML, IP, ASkalkidou, MJ, TP), methodology (ASvanberg, ML, IP, ASkalkidou, MJ, TP), formal analysis (JS, ASvanberg, FV, ASkalkidou, TP), investigation (JS, ASvanberg, FV, ML, MJ), data curation (all authors), supervision (ASvanberg, ML, TP), funding acquisition (ASvanberg, MJ), and project administration (ASvanberg, ML, IP, MJ). TP is the guarantor.

**Funding** This project was funded by The Regional Research Council (Regionala Forskningsrådet, RFR) grants nr 368901, 308451, and 480141; http://www.researchweb.org/ is/sverige) and Swedish research council funding for clinical research in medicine (ALF) grants nr N/A.

**Competing interests** None declared.

**Patient and public involvement** Patients and/or the public were not involved in the design, or conduct, or reporting, or dissemination plans of this research.

**Patient consent for publication** Not applicable.

**Ethics approval** The Regional Ethics Review Board in Uppsala, Sweden, approved the study (2012/495/1 and amendment 2016/11/16). Participants gave informed consent to participate in the study before taking part.

**Provenance and peer review** Not commissioned; externally peer reviewed.

**Data availability statement** Data are available upon reasonable request. Data sharing is available on reasonable request from researchers who provide a

methodologically sound proposal. Individual participant data that underlie the results reported in this article, after deidentification will be shared. Proposals should be directed to agneta.skoog_svanberg@kbh.uu.se. To gain access, data requestors will need to sign a data access agreement.

**ORCID iD**
Thomas Parling http://orcid.org/0000-0002-6159-598X

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
