## [Reviewer comments · BMJ Open]

ARTICLE DETAILS

TITLE (PROVISIONAL)	Antepartum and labour related single predictors of non-participation, dropout, and lost to follow up in a randomised controlled trial comparing internet-based cognitive behaviour therapy with treatment as usual for women with negative birth experiences and/or post-traumatic stress following childbirth.
AUTHORS	Sjömark, Josefin; Svanberg, Agneta; Viirman, Frida; Larsson, Margareta; Poromaa, Inger; Skalkidou, Alkistis; Jonsson, Maria; Parling, Thomas

VERSION 1 – REVIEW

REVIEWER	Biliunaite, Ieva Linköping University, Department of Behavioural Sciences and Learning
REVIEW RETURNED	02-May-2021

GENERAL COMMENTS	Thank you for the opportunity to review your manuscript. It is overall a well written article that discusses a very relevant and interesting topic. Below you will find my review comments attached. Abstract: Conclusion: I suggest including short reflection in relation to the clinical significance of the findings. (one or two sentences) Introduction: Overall, a good general overview. However, I suggest restructuring the introduction. Also, some important information is missing. My suggestions follow further. 1. I suggest including a short description of the ICBT in the first paragraph. Also, to provide with a couple of examples of psychological disorders that it has been used for. In relation to that, it could be useful to reflect on iCBT's effectiveness (shortly).2. 'In a PTSD study, the dropout rate in the iCBT group was 16%;' – could you please provide with more details? What was the study population etc.3. 'The form of an intervention can differ between different internet treatments and some patients need more support than what is offered to avoid dropout' – could you please shortly describe what is meant by more support and also, about the differing formats of the interventions (based on the context I assume that this is in relation to the therapist supported vs. self-help – if that is the case, please clarify).4. Please include a paragraph or two with some background information regarding your study population.
--

	5. I suggest placing the second last paragraph ('The literature is inconsistent regarding the definitions of participants who discontinue...') earlier in the introduction. 6. The last paragraph comes a bit abruptly, could you please link it/incorporate it into the whole a bit more smoothly? 7. I suggest including one paragraph where the main problem is summarized based on what was outlined in the text. That is, what is currently known and what is missing (that then leads to the main aim of this research work). Method: 1. Study design: it could be useful to define the four time points in here already. 2. Participants: 'For data over the eligible please see table 1.' – there is a word missing: Eligibility criteria or eligible participants? 3. Procedure: 'After giving birth and before departure from the hospital women rated their birth experience as a routine measure at the hospital' – was the experience rated twice? Please clarify shortly. 4. It is not clear from the text that out of 1223 participants 530 participants did not reply (only information about 693 is mentioned). Could you please state this in the text (I was only able to see this from the figure). 5. I would suggest including iCBT intervention details in the appendix; same regarding TAU – at least short summary of what it consists of. 6. 'Based on previous knowledge about possible causes for non-participation and dropout, predictor variables were categorized into three conceptual categories' – could be beneficial to list those here so that it is clear for the reader. 7. Statistical analysis: 'Missing data were not handled' – please shortly reflect on your motivation behind this decision. 8. Statistical analysis: Please mention that some of the information was categorized (results displayed in Table 3). Results: 1. Predictors of loss to follow up – in the text it is stated that the vaginal delivery predicted drop-out. In Table 2, immediate CS vs vaginal delivery and vaginal delivery vs vacuum delivery is highlighted as significant; please shortly reflect on this in text. 2. It might be beneficial to include some more information in the Table 2, such as, for example, beta values and their standard errors. Discussion: 1. 'The analysis included the inclusion criteria' paragraph – could you please reflect a bit more on the issue (such as your motivation to include the predictors etc). Currently this paragraph is very short and comes in a bit abruptly. Alternatively, this could be incorporated into a different paragraph. 2. Limitations: perception of birth experience and time point to measure it (as reflected in the discussion) could also be mentioned here. 3. I suggest adding a 'Clinical implications' paragraph or to elaborate more on the significance of the findings in the Conclusion paragraph.
--	--

REVIEWER	Rondung, Elisabet Mid Sweden University, Department of Psychology and Social Work
REVIEW RETURNED	25-May-2021

GENERAL COMMENTS	Thank you for giving me the opportunity to read a very interesting manuscript. These are my thoughts and comments on your work: ABSTRACT Lines 10-11: Is the word February lacking? If not, why specify the month in 2013 but not in 2018? Lines 10-12: From the abstract, it is not clear whether the 1523 women all were eligible or if it was women assessed for eligibility. Lines 18-19: Based on your use of the word “no” in this sentence, I have troubles understanding if you mean that absence or presence of severe anal sphincter injury and vaginal delivery predicted non-participation. Maybe you could rephrase it? Lines 24-25: Based on how you describe your results in the abstract, I don't see how you can conclude that foreign born, lower educated, multipara mothers were more likely to drop-out from the intervention? Non-participate is ok, but drop out? INTRODUCTION I find your introduction easy to follow and relevant in relation to the topic. Lines 6:24-7:1. On lines 6:14-16 you refer to Melville's three categories of dropout predictors. How come you have not included psychological problems and treatment-related variables in your objective? At least for treatment drop out you should have access to these? If you do not have the possibility to include these (which I would suggest you do), why then present this model? METHODS Lines 7:13-16: Would you call this a longitudinal design? Line 7:20: Could it be possible to give the number of women that rated their overall birth experience in parenthesis? Lines 7:22-23: Are the data that guide inclusion retrieved from medical records? Table 1: Is it necessary to include age in the table or could it be reported in text only, in order to avoid mixing the M (SD) and n (%) formats in the table? Later on, when trying to interpret your findings I realize that I miss having the numbers of women categorized according to you predictor variables in each of your subsamples. It is difficult to understand for example the burden of severe hemorrhage when you (as a reader) do not know the number of women that experienced this complication. Figure 1: How many of the women that were lost to follow-up were from the treatment and the TAU-groups respectively? How large was the overlap between the groups Treatment dropouts and Lost to follow-up? RESULTS Page numbers from 10 and onward need to be checked (all 1).
---

	Table 2: I find it difficult to understand how to read some of the predictors, e.g. the variables for mode of delivery. Why not use one (vaginal delivery) as a reference, and state OR 1.0 and then list the others below (and skip the vs.)? Why do you think no severe postpartum hemorrhage predicted pre-treatment dropout? And why would no intrapartum fetal distress predict loss to follow up? Why would fetal presentation predict treatment drop out but no other outcomes? And why would intrapartum fetal distress predict loss to follow-up but not drop out? When I look at Table 2, I find it difficult to understand the pattern of significant predictors. Is it possible that your alpha level ($p < .05$) is too low given the large number of predictors in your analyses? Or could it be the varying numbers of women in each subsample? Are these variables more or less dominant in some of the samples? Since you don't present the number of women represented by each category of the predictor variables this is difficult to know. If you want to draw attention to these findings, I would suggest that you present the n for each category and elaborate your discussion of these findings. Could treatment group be a significant predictor of loss to follow up? I think that would be an important predictor to include. DISCUSSION Lines 13: 8-9: I am not sure I would agree that usage of the Likert scale (or VAS-scale) is well-established in research. It might be so in clinical care, at least in Sweden, but do you really find it well-established in research? Lines 13: 16-19: You discuss the fact that you used your inclusion criteria as predictors. I agree that this really needs to be interpreted with caution. When you write that "the results might be useful for future hypothesis in further research", do you have any more specific ideas about the implementation of this finding? Lines 14:4-22. I think it could be a good idea to integrate your findings with your thoughts about why the significant predictors are associated with the outcomes. For example, you write "Childbirth without large hemorrhage and a less severe overall birth experience predicted pre-treatment dropout, while vertex fetal presentation and vaginal delivery predicted treatment dropout", but you don't reveal any of your thoughts about this. In the following section you start with stating another association "Predictors for loss to follow-up were vaginal delivery vs. instrumental delivery" but the discussion that follows does not match that finding. Instead you describe some of your predictors, but not in relation to the variables you have shown an association with. I'm sure you have interesting thoughts about why these associations have been found, but at the moment, these are not clearly presented. Conclusion Your conclusion makes an important contribution to your paper. Until I reached the conclusion I thought you were a bit unexplicit (e.g. on lines 14:23-15:3 and page 15, lines 11-15). Lines 15-16: "In summary, both demographic and obstetrical variables are important to attend to while designing procedures to maximize participation in iCBT." Yes, if conducting studies of postpartum (and perhaps also pregnant) women.
--	---

VERSION 1 – AUTHOR RESPONSE

Reviewer: 1

Ms. Ieva Biliunaite, Linköping University

Comments to the Author:

Thank you for the opportunity to review your manuscript. It is overall a well written article that discusses a very relevant and interesting topic. Below you will find my review comments attached.

Authors: Thank you

Abstract:

Conclusion: I suggest including short reflection in relation to the clinical significance of the findings. (one or two sentences)

Reply: Thank you, we have added “Both demographic, antepartum and obstetrical variables are important to attend to while designing procedures to maximize participation in iCBT.” in the conclusions in the Abstract

Introduction:

Overall, a good general overview. However, I suggest restructuring the introduction. Also, some important information is missing. My suggestions follow further.

1. I suggest including a short description of the iCBT in the first paragraph. Also, to provide with a couple of examples of psychological disorders that it has been used for. In relation to that, it could be useful to reflect on iCBT's effectiveness (shortly).

Reply: We have added the following in the first paragraph “The active mechanisms in iCBT are the same as in CBT but it differs in the way it is delivered (internet-/computer-based) and increases the availability for evidence based psychological interventions in the society. iCBT is convenient, flexible, and cost-effective for many different psychological disorders (2); it is effective for treatment of depression and several anxiety disorders, and for some diagnoses, iCBT is equally effective as face-to-face CBT (3,4).”

2. 'In a PTSD study, the dropout rate in the iCBT group was 16%;' – could you please provide with more details? What was the study population etc.

Reply: We have added the following “In a study 96 adult patients with posttraumatic stress reactions were allocated to ten sessions of iCBT or waiting list control. The dropout rate in the iCBT group was 16%; technical problems and emotional distress due to the treatment interventions were the most frequently reported dropout reasons (20).”

3. 'The form of an intervention can differ between different internet treatments and some patients need more support than what is offered to avoid dropout' – could you please shortly describe what is meant by more support and also, about the differing formats of the interventions (based on the context I assume that this is in relation to the therapist supported vs. self-help – if that is the case, please clarify).

Reply: We have added the following “The form of an intervention can differ between different internet treatments, considering amount of material, intensity and support. Some interventions are, e.g., fully therapist-supported with face-to-face sessions or via phone, some offer support via mail, and some do not offer support at all (self-help) (2).”

4. Please include a paragraph or two with some background information regarding your study population.

Reply: We have added this information in the paragraph above declaration of the aims of the study. The following is added. "About 115 000 women give birth in Sweden every year (Socialstyrelsen). Childbirth is a subjective and multidimensional event that in some cases can lead to a negative childbirth experience. The prevalence of negative childbirth experiences varies (9-45%) in different communities (22-24). For some women (3-4%) the distress of a negative childbirth experience lead to the development of Posttraumatic Stress Disorder Following Childbirth (PTSD FC) (25-30). In Sweden there is no specific treatment recommendation for women with negative birth experiences and or PTSD FC. So far, only a few randomised and controlled trials have investigated the efficacy of different interventions for this population, it is therefore none or little information about how women with negative birth experiences commit and engage in iCBT and similar treatments."

5. I suggest placing the second last paragraph (' The literature is inconsistent regarding the definitions of participants who discontinue...') earlier in the introduction.

Reply: Thank you, this is a very helpful suggestion, this paragraph is now earlier in the introduction. We think that this suggestion made the introduction much better.

6. The last paragraph comes a bit abruptly, could you please link it/incorporate it into the whole a bit more smoothly?

Reply: See point 4. This addition provides a smoother link to the aim of the study. Thank you for this suggestion.

7. I suggest including one paragraph where the main problem is summarized based on what was outlined in the text. That is, what is currently know and what is missing (that then leads to the main aim of this research work).

Reply: Thanks again, see point 4 and 6, this is now added.

Method:

1. Study design: it could be useful to define the four time points in here already.

Reply: This is added now under Study design. "Investigation of single predictors for non-participation, pre-treatment dropout, treatment dropout and loss to follow up, reflecting four consecutive time points (about 8 weeks postpartum, about 10 weeks postpartum, between 10 and 16 weeks postpartum, and after 16 weeks postpartum respectively), for all eligible participants in a longitudinal RCT."

2. Participants: 'For data over the eligible please see table 1.' – there is a word missing: Eligibility criteria or eligible participants?

Reply: Thank you, eligible participants is now added

3. Procedure: ' After giving birth and before departure from the hospital women rated their birth experience as a routine measure at the hospital' – was the experience rated twice? Please clarify shortly.

Reply: The sentence is changed to (Under Procedure): Women rated their birth experience as a routine measure at the hospital before discharge.

4. It is not clear from the text that out of 1223 participants 530 participants did not reply (only information about 693 is mentioned). Could you please state this in the text (I was only able to see this from the figure).

Reply: This Information is now added (under Procedure): "In total 530 eligible women did not respond to the invitation, 300 women"

5. I would suggest including iCBT intervention details in the appendix; same regarding TAU – at least short summary of what it consists of.

Reply: we have added a table suitable as an appendix for the iCBT content. We have no further data on what TAU consists of exactly, we think that the description of TAU is as detailed as it can be. However, we want to be as clear as possible and are grateful for input on what specifically could be added.

6. 'Based on previous knowledge about possible causes for non-participation and dropout, predictor variables were categorized into three conceptual categories' – could be beneficial to list those here so that it is clear for the reader.

Reply: The categories are added (under Material heading): "Based on previous knowledge about possible causes for non-participation and dropout, predictor variables were categorized into three conceptual categories (demographic, antepartum, and labour-related/postpartum variables)."

7. Statistical analysis: 'Missing data were not handled' – please shortly reflect on your motivation behind this decision.

Reply: Among the predictor variables there were 3 demographics that had missing data above 5%; country of birth, education, and relationship status (18,5%, 12%, and 10,8% missing respectively). We decided not to impute these demographic missing data from antepartum, and labour-related/postpartum variables, as we think it would have resulted in arbitrary imputations. In addition, since the analyses are not multivariate but bivariate the consequences are less.

8. Statistical analysis: Please mention that some of the information was categorized (results displayed in Table 3).

Reply: This is added (Statistical analysis): Reasons given for non-participation were categorized.

Results:

1. Predictors of loss to follow up – in the text it is stated that the vaginal delivery predicted drop-out. In Table 2, immediate CS vs vaginal delivery and vaginal delivery vs vacuum delivery is highlighted as significant; please shortly reflect on this in text.

Reply: Yes thank you, we have also added the following in the results section, loss to follow up: "In the analyses of loss to follow-up, absence of intrapartum fetal distress and vaginal delivery (compared with immediate CS and vacuum delivery) predicted loss to follow up (Table 2)."

2. It might be beneficial to include some more information in the Table 2, such as, for example, beta values and their standard errors.

Reply: Thank you. We have added beta values and SE, as well as number of participants in each subsample for clarity in Table 2. The information in Table 2 might get a bit overloaded due to the beta values and SE. In addition, beta values are less informative / pedagogical / intuitive when it comes to categorical variables (with continuous data it is very informative at first sight though). All but 2 variables are categorical. We leave it up to the editor and reviewer to decide if the beta values and SE information shall be presented. Or if it should be presented in an appendix. As of now beta values SE are added in Table 2.

Discussion:

1. 'The analysis included the inclusion criteria' paragraph – could you please reflect a bit more on the issue (such as your motivation to include the predictors etc). Currently this paragraph is very short and come in a bit abruptly. Alternatively, this could be incorporated into a different paragraph.

Reply: Thank you. We have added a section regarding the inclusion criteria as predictors. "The analysis included the inclusion criteria (immediate cesarean section, overall birth experience, and severe hemorrhage) as predictors. Non-participation was predicted by vaginal delivery vs. immediate cesarean section. Childbirth without severe hemorrhage predicted pre-treatment dropout. It is known that severe postpartum hemorrhage is a significant risk factor for developing PTSD (27,28). Treatment drop out was predicted by a less severe overall birth experience and vaginal delivery vs. immediate CS. These predictors were inclusion criteria and should be interpreted with caution as mentioned above. The results regarding these predictors must therefore be interpreted with caution. However, the results might be useful for future hypothesis in further research. The three inclusion criteria in this study are experiences that potentially can have serious effects on the mental health of a birth giving woman. It may be of value to understand more about what type of care (e.g., counselling, therapy), what type of format (e.g., face to face or ICBT) and what level of support (therapist support or pure self-help) is demanded. "

2. Limitations: perception of birth experience and time point to measure it (as reflected in the discussion) could also be mentioned here.

Reply: We have added the following "Further, recruitment to the study was before discharge from the hospital when the experience of birth is fresh. Thus, the eligible sample might have been different if we had asked them at a later time point. However, the assessment of the birth experience was in close conjunction to immediate CS and severe hemorrhage (the two other inclusion criteria)."

3. I suggest adding a 'Clinical implications' paragraph or to elaborate more on the significance of the findings in the Conclusion paragraph.

Reply: Thank you, we have added the following trying to balance brevity and clarity "In summary, both demographic and obstetrical variables are important to attend to for both clinical and research purposes, while designing procedures to maximize participation in iCBT for postpartum women."

Reviewer: 2

Dr. Elisabet Rondung, Mid Sweden University

Comments to the Author:

Thank you for giving me the opportunity to read a very interesting manuscript. These are my thoughts and comments on your work:

ABSTRACT

Lines 10-11: Is the word February lacking? If not, why specify the month in 2013 but not in 2018?

Reply: thanks this is added

Lines 10-12: From the abstract, it is not clear whether the 1523 women all were eligible or if it was women assessed for eligibility.

Reply: the 1523 is the sample and they are the eligible.

Lines 18-19: Based on your use of the word "no" in this sentence, I have troubles understanding if you mean that absence or presence of severe anal sphincter injury and vaginal delivery predicted non-participation. Maybe you could rephrase it?

Reply: Yes, to be more clear we changed it: Non-participation was predicted by lower level of education, multiparity, being foreign-born, absence of preeclampsia, absence of severe anal sphincter injury, no experience of counseling for fear of childbirth and vaginal delivery (vs. cesarean section and vacuum assisted delivery).

Lines 24-25: Based on how you describe your results in the abstract, I don't see how you can

conclude that foreign born, lower educated, multipara mothers were more likely to drop-out from the intervention? Non-participate is ok, but drop out?

Reply: We added the following in the results section in the abstract.

“Treatment dropout was predicted by vaginal delivery (compared with immediate cesarean section), vertex presentation and good overall birth experience.”

We also changed the conclusion in the abstract to be more general: “Mothers with no obstetric complications were more likely to not participate and dropout at different time points from the internet intervention. Both demographic, antepartum and obstetrical variables are important to attend to while designing procedures to maximize participation in iCBT.”

INTRODUCTION

I find your introduction easy to follow and relevant in relation to the topic.

Reply: Thank you.

Lines 6:24-7:1. On lines 6:14-16 you refer to Melville’s three categories of dropout predictors. How come you have not included psychological problems and treatment-related variables in your objective? At least for treatment drop out you should have access to these? If you do not have the possibility to include these (which I would suggest you do), why then present this model?

Reply: In this study we chose to focus on birth related variables and they are assessed / collected at the time of the birth. We wanted the predictor variables to be assessed / collected at the same time point. Psychological problems at this time point were not assessed or not written in the patient’s medical records for a majority of our participants. We did not include psychological problems variables due to that. We still think that the model is of value due to the heterogenous use and definition of the terms in research. Melville’s identification and definition of these terms are of good value for further research in this topic.

METHODS

Lines 7:13-16: Would you call this a longitudinal design?

Reply: Yes we have added that the RCT is a longitudinal study and since the predictor variables are collected before non-participation, pre-treatment drop out, treatment dropout and loss of follow-up this study is also longitudinal.

Line 7:20: Could it be possible to give the number of women that rated their overall birth experience in parenthesis?

Reply: This is now added under Participants, in the second sentence “Approximately 17,000 women gave birth at anonymised Hospital between September 2013 and February 2018, and a majority (n = 1,203) rated their overall birth experience on a Likert scale”

Lines 7:22-23: Are the data that guide inclusion retrieved from medical records?

Reply: Yes they are.

Table 1: Is it necessary to include age in the table or could it be reported in text only, in order to avoid mixing the M (SD) and n (%) formats in the table?

Reply: Age is now in text under participants, this is added “The 1,523 eligible women had a mean age of 31.5 years (SD = 5.03), participants in the RCT study were 31.7 (4.6) years, and the non-participants age were 31.4 (5.1) years; the majority reported.....”

Later on, when trying to interpret your findings I realize that I miss having the numbers of women categorized according to your predictor variables in each of your subsamples. It is difficult to understand for example the burden of severe hemorrhage when you (as a reader) do not know the number of women that experienced this complication.

Reply: Thank you, please see table 2, this is now added for each subsample.

Figure 1: How many of the women that were lost to follow-up were from the treatment and the TAU-groups respectively? How large was the overlap between the groups Treatment dropouts and Lost to follow-up?

Reply: In figure 1 we added "(TAU n=32, ICBT+TAU n=46)" in the lost to follow up. N=35 of the treatment dropouts were lost to follow up (total n=58).

RESULTS

Page numbers from 10 and onward need to be checked (all 1).

Reply: So sorry, must have been something due to the transferring system, the document looks good on our end.

Table 2: I find it difficult to understand how to read some of the predictors, e.g. the variables for mode of delivery. Why not use one (vaginal delivery) as a reference, and state OR 1.0 and then list the others below (and skip the vs.)?

Reply: Thank you, a very good suggestion, We have done that. Please see table 2.

Why do you think no severe postpartum hemorrhage predicted pre-treatment dropout?

Reply: We have added the following regarding the predictor variables that were also inclusion criteria "The analysis included the inclusion criteria (immediate cesarean section, overall birth experience, and severe hemorrhage) as predictors. Non-participation was predicted by vaginal delivery vs. immediate cesarean section. Childbirth without severe hemorrhage predicted pre-treatment dropout. It is known that severe postpartum hemorrhage is a significant risk factor for developing PTSD (27,28). Treatment drop out was predicted by a less severe overall birth experience and vaginal delivery vs. immediate CS. These predictors were inclusion criteria and should be interpreted with caution as mentioned above." Although these findings (inclusion criteria) are consistent with the rest of the significant predictors we think it is better to not elaborate more on these findings. Further research can hopefully investigate them.

And why would no intrapartum fetal distress predict loss to follow up?

Reply: We have added the following "Predictors for loss to follow-up were vaginal delivery vs. instrumental delivery and absence of intrapartum fetal distress. As mentioned above, the occurrence of these events are threats to the fetus which in turn can be very stressful for the mother. Absence of these events might lead to lost interest in devoting time and energy to proceed with the follow-up assessment. Absence of immediate CS was also a significant predictor of loss to follow up and is discussed in relation to the other inclusion criteria predictors. "

Why would fetal presentation predict treatment drop out but no other outcomes?

Reply: The analyses regarding treatment drop out have the least power due to the relatively few participants. In table 2 we have added the n for each subsample for further inspection for the reader.

And why would intrapartum fetal distress predict loss to follow-up but not drop out?

Reply: This might be a result depending on the number of participants that differ in the two analyses. We do present the finding but chose not to discuss that discrepancy. It is probably due to the larger n in the loss to follow up analyses.

When I look at Table 2, I find it difficult to understand the pattern of significant predictors. Is it possible that your alpha level ($p < .05$) is too low given the large number of predictors in your analyses? Or could it be the varying numbers of women in each subsample?

Reply: We have now added information on n for each subsample. These predictors are variables that were based on previous research on register studies and as such decided upon as important in the planning of the study. Therefore, we did not use e.g. Bonferroni correction. If the editor or reviewer want us to add such a correction we will.

Are these variables more or less dominant in some of the samples? Since you don't present the number of women represented by each category of the predictor variables this is difficult to know. If you want to draw attention to these findings, I would suggest that you present the n for each category and elaborate your discussion of these findings.

Reply: The number of each subsample is now added in table 2.

Could treatment group be a significant predictor of loss to follow up? I think that would be an important predictor to include.

Reply: We have added treatment group as a predictor for loss to follow up, the following is added in the Results section: "An additional analysis showed that being randomised to iCBT+TAU was a significant predictor of loss to follow up OR = 1.84 (95 % CI: 1.04-3.28), B=0.61, SE=0.29, $p = .037$, where 46 of 99 in iCBT+TAU and 32 of 100 in TAU were lost to follow up." We also added a short reflection in the discussion in the section regarding predictors for loss to follow up: "Finally, randomisation to iCBT+TAU (compared with TAU) was a significant predictor of lost to follow up. A majority of those who were lost to follow up from the iCBT+TAU group were also those who were treatment dropouts."

DISCUSSION

Lines 13: 8-9: I am not sure I would agree that usage of the Likert scale (or VAS-scale) is well-established in research. It might be so in clinical care, at least in Sweden, but do you really find it well-established in research?

Reply: We have changed that sentence into "However, using a Likert scale as a tool for self-assessment of overall birth experience is well-established in clinical practice and used in research (22,23)."

Lines 13: 16-19: You discuss the fact that you used your inclusion criteria as predictors. I agree that this really needs to be interpreted with caution. When you write that "the results might be useful for future hypothesis in further research", do you have any more specific ideas about the implementation of this finding?

Reply: We have added some discussion regarding the inclusion criteria and their role as predictors. Your comment is warranted. The discussion on the inclusion criteria predictors is now at one place and it will help the reader to gain some more understanding on our position regarding these. We have added the following: "The analysis included the inclusion criteria (immediate cesarean section, overall birth experience, and severe hemorrhage) as predictors. Non-participation was predicted by vaginal delivery vs. immediate cesarean section. Childbirth without severe hemorrhage predicted pre-treatment dropout. It is known that severe postpartum hemorrhage is a significant risk factor for developing PTSD (27,28). Treatment drop out was predicted by a less severe overall birth experience and vaginal delivery vs. immediate CS. These predictors were inclusion criteria and should be interpreted with caution as mentioned above. The results regarding these predictors must therefore be interpreted with caution. However, the results might be useful for future hypothesis in further research. The three inclusion criteria in this study are experiences that potentially can have serious effects on the mental health of a birth giving woman. It may be of value to understand more about

what type of care (e.g., counselling, therapy), what type of format (e.g., face to face or ICBT) and what level of support (therapist support or pure self-help) is demanded.”

Lines 14:4-22. I think it could be a good idea to integrate your findings with your thoughts about why the significant predictors are associated with the outcomes. For example, you write “Childbirth without large hemorrhage and a less severe overall birth experience predicted pre-treatment dropout, while vertex fetal presentation and vaginal delivery predicted treatment dropout”, but you don’t reveal any of your thoughts about this.

Reply: We moved the discussion regarding inclusion criteria as predictors to one section, please see the above reply.

We also added the following sentence regarding the antepartum and labor related / postpartum variables that predicted non-participation “The other significant labor related / post-partum predictors show a consistent pattern where the women who did not experience these stressful events may not have had enough motivation to seek out help or support.”

Vertex presentation is now discussed. We added the following “Vertex fetal presentation (vs. other presentation) predicted treatment dropout. This is consistent with the significant predictors for non-participation where those with vertex presentation might not experience this as a stressful event enough to stay in the treatment. It might also be that those with vertex presentation that were randomized to the treatment did not find it helpful or that it did not address their problem fully to stay in the treatment.”

In the following section you start with stating another association “Predictors for loss to follow-up were vaginal delivery vs. instrumental delivery” but the discussion that follows does not match that finding. Instead you describe some of your predictors, but not in relation to the variables you have shown an association with. I’m sure you have interesting thoughts about why these associations have been found, but at the moment, these are not clearly presented.

Reply: Thank you. We moved the section that did not match the variables that were presented, up to the non-participation discussion where they should have been. We added the following to the discussion of predictors for loss to follow up “Predictors for loss to follow-up were vaginal delivery vs. instrumental delivery and absence of intrapartum fetal distress. As mentioned above, the occurrence of these events are threats to the fetus which in turn can be very stressful for the mother. Absence of these events might lead to lost interest in devoting time and energy to proceed with the follow-up assessment. Absence of immediate CS was also a significant predictor of loss to follow up and is discussed in relation to the other inclusion criteria predictors.”

Conclusion

Your conclusion make an important contribution to your paper. Until I reached the conclusion I thought you were a bit unexplicit (e.g. on lines 14:23-15:3 and page 15, lines 11-15).

Reply: Thank you.

Lines 15-16: “In summary, both demographic and obstetrical variables are important to attend to while designing procedures to maximize participation in iCBT.” Yes, if conducting studies of postpartum (and perhaps also pregnant) women.

Reply: Yes, this is added.

VERSION 2 – REVIEW

REVIEWER	Rondung, Elisabet Mid Sweden University, Department of Psychology and Social Work
REVIEW RETURNED	05-Apr-2022

GENERAL COMMENTS	Dear authors, It has been a pleasure to read your responses and the updated version of the manuscript. Great work! I have just a few more, minor, comments: In the participants section, I don't really get how 1,203 can be the majority of 17,000. I guess there is something missing here - maybe the number of women rating on the VAS scale? In table 2, I appreciate the numbers (N) included and when you show your reference clearly on a specific row (OR = 1.0). However, the variables where you do not show the reference like this, I find a bit ambiguous. When you write for example yes/no it is not (at least to me) crystal clear which is the reference and which is the one for which you present the odds ratio. Could this be clarified? Would it also be possible to clarify that a few variables are continuous, in order to make interpretation more straight forward? In the discussion you write "These predictors were inclusion criteria and should be interpreted with caution as mentioned above. The results regarding these predictors must therefore be interpreted with caution." Maybe this could be formulated without repeating "interpreted with caution"? As you discuss in the response, the varying N in different analyses might have had some importance for which predictors came out as significant or not. Would it be possible to include a short discussion relating to this in the limitations section? With the revisions made and after considering these suggestions, I'd be happy to see your manuscript published. Best wishes, Elisabet
--

VERSION 2 – AUTHOR RESPONSE

Reviewer: 1
Dr. Elisabet Rondung, Mid Sweden University

Comments to the Author:

Dear authors,
It has been a pleasure to read your responses and the updated version of the manuscript. Great work!
I have just a few more, minor, comments:

- In the participants section, I don't really get how 1,203 can be the majority of 17,000. I guess there is something missing here - maybe the number of women rating on the VAS scale?

Reply: Thank you for seeing that mistake, the number 1,203 should not be there at all, we now changed the text to:

Approximately 17,000 women gave birth at Uppsala university Hospital between September 2013 and February 2018, and most of them rated their overall birth experience on a Likert scale (0–10), as a standard procedure before hospital discharge.

- In table 2, I appreciate the numbers (N) included and when you show your reference clearly on a specific row (OR = 1.0). However, the variables where you do not show the reference like this, I find a bit ambiguous. When you write for example yes/no it is not (at least to me) crystal clear which is the reference and which is the one for which you present the odds ratio. Could this be clarified? Would it also be possible to clarify that a few variables are continuous, in order to make interpretation more straight forward?

Reply: Yes, we agree, we have now inserted a note under table 2:

Note. The first category is the reference, for e.g. when yes/no is stated, yes is the reference category.

- In the discussion you write "These predictors were inclusion criteria and should be interpreted with caution as mentioned above. The results regarding these predictors must therefore be interpreted with caution." Maybe this could be formulated without repeating "interpreted with caution"?

Reply: Thank you for seeing that being repeated! We now changed the sentence to:

These predictors were inclusion criteria and must therefore be interpreted with caution. However, the results might be useful for future hypothesis in further research.

- As you discuss in the response, the varying N in different analyses might have had some importance for which predictors came out as significant or not. Would it be possible to include a short discussion relating to this in the limitations section?

Reply: Thank you, that is an important thing to address, we inserted the following sentence in the limitations section:

In some analyses there might have been a lack of power, due to the varying N, that prevented significant predictors to be found.

With the revisions made and after considering these suggestions, I'd be happy to see your manuscript published.

Reply: Thank you very much, again for helping us improve our manuscript!

Best wishes,
Elisabet

VERSION 3 – REVIEW

REVIEWER	Rondung, Elisabet Mid Sweden University, Department of Psychology and Social Work
REVIEW RETURNED	26-Aug-2022
GENERAL COMMENTS	Thank you for giving me the opportunity to read your manuscript once more. I am completely satisfied with how you have responded to my comments and wish you the best of luck! / Elisabet